

# Temperament and character effects on late adolescents' well-being and emotional-behavioural difficulties

Cristiano Crescentini[1], Marco Garzitto[2], Andrea Paschetto[3], Paolo Brambilla[4,5] and Franco Fabbro[6,7]

[1] Department of Languages and Literatures, Communication, Education and Society, University of Udine, Udine, Italy
[2] Scientific Institute, IRCCS Eugenio Medea, San Vito al Tagliamento, Pordenone, Italy
[3] ISIS "E. Mattei", Latisana/Lignano Sabbiadoro, Udine, Italy
[4] Scientific Institute, IRCCS Eugenio Medea, Bosisio Parini, Lecco, Italy
[5] Department of Pathophysiology and Transplantation, University of Milan, Milan, Italy
[6] Department of Medicine, University of Udine, Udine, Italy
[7] Scuola Superiore Sant'Anna, Perceptual Robotics (PERCRO) Laboratory, Pisa, Italy

## ABSTRACT

**Background**. Research on adults points to personality as a crucial determinant of well-being. The present study investigates the question of personality's relation to well-being and psychosocial adjustment in adolescence.

**Methods**. We assessed the role of temperament and character (Temperament and Character Inventory, TCI-125), on psychological well-being (PWB; Psychological Well-Being scales), subjective well-being (SWB; Positive and Negative Affect, PA and NA, respectively), and psychosocial adjustment (emotional-behavioural problems measured by the Strengths and Difficulties Questionnaire for Adolescents, SDQ-A), in 72 Italian late adolescents (aged $17.5 \pm 0.75$). Multiple regressions were conducted to predict PWB, SWB, and SDQ-A scores using TCI-125 scales as predictors.

**Results**. Character maturity, and in particular Self-Directedness, had a widespread protective effect on well-being and psychosocial adjustment, while different strengths and emotional-behavioural difficulties were associated to specific temperamental and character traits. For example, Harm-Avoidance and Novelty-Seeking positively predicted internalized and externalized problems, respectively.

**Discussion**. The present results suggest the usefulness of continuing to evaluate temperament and, in particular, character dimensions in investigations focused on adolescents' well-being and psychosocial functioning, especially in the contexts of potential interventions aimed at enhancing development of adolescents' character dimensions at the intrapersonal, interpersonal, and transpersonal levels.

# INTRODUCTION

Research on adults and adolescents points to personality as an underlying core factor influencing well-being (*Schmutte & Ryff, 1997*; *Fogle, Huebner & Laughlin, 2002*;

Corresponding author
Cristiano Crescentini,
cristiano.crescentini@uniud.it

*Cloninger & Zohar, 2011*; *Butkovic, Brkovic & Bratko, 2012*; *Moreira et al., 2015*). This research is largely focused on trait models of personality. Thus, in the context of the Big-Five personality Inventory (BFI; *Costa & McCrae, 1992*) and the Eysenck Personality Questionnaire (*Eysenck & Eysenck, 1975*), distinct but related aspects of well-being, such as hedonic, subjective well-being (SWB; measured by: life satisfaction; positive affect, PA; negative affect, NA; and happiness and referring to the affective dimensions of an individual's life experiences) and eudaimonic, psychological well-being (PWB; indexed by: autonomy, personal growth, self-acceptance, purpose in life, environmental mastery, and positive relations with others in the model of *Ryff, 1989*) are associated with adults and adolescents' personality traits. For example, neuroticism is negatively associated with happiness, self-acceptance, and environmental mastery and positively with NA; extraversion is positively associated with PA, happiness, life satisfaction, and personal growth; also conscientiousness and agreeableness appear to be positively related with life satisfaction, PA, and aspects of PWB (e.g., *Schmutte & Ryff, 1997*; *Steel, Schmidt & Shultz, 2008*; *Garcia, 2011*; *Butkovic, Brkovic & Bratko, 2012*; see also *Moreira et al., 2015*).

A series of important recent studies helps to reconsider the investigation of the relationship between personality and well-being by taking into account the interplay between heritable personality traits and their interaction with environment and learning, which dynamically and flexibly shape individuals' concepts of the self, intentional values and goals, namely the character. After the seminal work on adults of Park, Peterson, and Seligman (*Park, Peterson & Seligman, 2004*), several longitudinal and cross-sectional studies have investigated the relationship between adolescents' character strength/maturity and well-being (*Garcia, 2011*; *Gillham et al., 2011*; *Garcia & Moradi, 2012*; *Garcia et al., 2012*; *Garcia, Kerekes & Archer, 2012*; *Nima, Archer & Garcia, 2012*; *Schültz, Archer & Garcia, 2013*; *Moreira et al., 2015*), often employing the psychobiological model of personality developed by Cloninger and colleagues (the Temperament and Character Inventory, TCI; *Cloninger, Svrakic & Przybeck, 1993*).

In this latter model, personality is organized into temperament and character dimensions. Based on a neurobiological model of behavioural conditioning, temperament refers to individual differences in reactivity, intensity and duration of responses to basic emotional stimuli pertaining to anger, fear, disgust, and ambition. Four TCI temperament dimensions are considered: Novelty Seeking (NS; e.g., exploratory and impulsive vs indifferent and reflective); Harm Avoidance (HA; e.g., worrying and anxious vs relaxed and confident); Reward Dependence (RD; e.g., sentimental and dependent vs practical and independent), and Persistence (Pe; e.g., industrious and perseverant vs inactive and quitting). Character refers to individual differences in higher-order propositional or insight learning of self-concepts, intentional values, and personal goals. In the TCI, the maturity of the self, i.e., the character, is defined at three levels: at the intrapersonal level with Self-Directedness (SD; purposeful, responsible and reliable vs purposeless, blaming and unreliable), at the interpersonal level with Cooperativeness (Co; helpful, empathic and ethical vs unhelpful, critical and opportunistic), and at the transpersonal level with Self-Transcendence (ST; holistic and united with the universe vs self-centered and unimaginative) (*Cloninger, Svrakic & Przybeck, 1993*; *Cloninger, 2004*). It was suggested

(*Cloninger, Svrakic & Przybeck, 1993*) and validated (*Svrakic et al., 1993*; *Cloninger & Zohar, 2011*; *Josefsson et al., 2011*) that a mature character mostly develops on intrapersonal and interpersonal dimensions, so that the combination of high SD and Co scores is indicative of character maturity and a protective factor against personality disorder (*Cloninger, Svrakic & Przybeck, 1993*; *Kluger et al., 1999*).

Notably, most of the studies on the relationship between character maturity and well-being in adolescents focuses on SWB (*Gillham et al., 2011*; *Garcia & Moradi, 2012*; *Garcia, Kerekes & Archer, 2012*; *Nima, Archer & Garcia, 2012*; *Schültz, Archer & Garcia, 2013*), with only a minority of recent studies also including PWB (*Garcia, 2011*; *Moreira et al., 2015*). This is important if one considers that the character could be particularly associated to PWB, which refers to the congruence between life experiences and values and goals of an individual (*Moreira et al., 2015*; *Cloninger, 2004*).

All these studies show how complex are the relationships between adolescents' personality and well-being. From another perspective, several investigations on adults and adolescents also emphasize that well-being may not be simply the absence of symptomatology or distress, as suggested by studies showing that removal of individuals' distress does not inevitably lead to engendering well-being (see 'Discussion' in *Ruini et al., 2003b*; *Vescovelli, Albieri & Ruini, 2014*; *Singh et al., 2015*). Nevertheless, well-being and distress may not be mutually exclusive, but rather coexisting dimensions, in adults as well as in adolescents (*Vescovelli, Albieri & Ruini, 2014*), as documented by other findings showing less behavioural, social, and emotional problems in adolescents with high levels of well-being (*Park, Peterson & Seligman, 2004*; *Suldo & Huebner, 2006*; *Singh et al., 2015*). Notwithstanding, for a complete conception of optimal individual functioning in adolescence, it appears crucial to consider both the concept of well-being, and its relationship to personality, and that of psychosocial adjustment, again considering the associations with personality.

Although the frequency of several forms of psychopathology, including behavioural and affective disorders, significantly increases during adolescence (*Silk, Steinberg & Morris, 2003*), the relationship between positive personality traits, such as character strengths, and decreased risk of developing psychopathology symptoms is scarcely explored in adolescents. As for well-being, many of the cross-sectional and longitudinal studies on the relation between personality and maladjustment and psychopathology indices in adolescence focus on traits models of personality and temperament dimensions (*Barnow, Lucht & Freyberger, 2005*; *Muris, Meesters & Diederen, 2005*; *Suldo & Huebner, 2006*; *Hilt, Armstrong & Essex, 2012*). For example, it is documented that agreeableness and conscientiousness are negatively related to conduct problems and symptoms of inattention and hyperactivity (measured by the Strengths and Difficulties Questionnaire, SDQ; *Goodman, 1997*; *Goodman, 2001*), while neuroticism is more globally positively associated with all difficulties measured by the SDQ (*Muris, Meesters & Diederen, 2005*).

Finally, a restricted number of studies focusing on character strengths, both within (*Garcia et al., 2012*) and outside the TCI framework (*Bromley, Johnson & Cohen, 2006*; *Gillham et al., 2011*), shows that adolescents who report high levels of persistence and character maturity (high scores in SD and Co; *Garcia et al., 2012*), or other character

strengths (e.g., forgiveness, kindness, self-regulation, perseverance, productive activity, confident optimism; *Bromley, Johnson & Cohen, 2006*; *Gillham et al., 2011*), report low levels of depressive symptoms (*Gillham et al., 2011*; *Garcia et al., 2012*) or decreased risks of developing psychiatric disorders during early adulthood (*Bromley, Johnson & Cohen, 2006*). Overall, the lack of systematic investigation of PWB and SWB in these studies makes it difficult to ascertain in the same subjects potential similarities with the effects of personality strength on psychosocial functioning.

The present study investigates the question of temperament and character's relation to well-being and psychosocial adjustment in adolescence. A group of Italian late adolescents self-reported personality in the Cloninger's psychobiological model of personality (TCI). Moreover, they reported well-being, measured both as PWB (*Ryff, 1989*) and SWB (*Watson, Clark & Tellegen, 1988*), and psychosocial adjustment (i.e., emotional-behavioural difficulties; *Goodman, 1997*; *Goodman, 2001*). Based on previous research, our hypothesis was that temperament and character dimensions may significantly predict different aspects of well-being and psychosocial functioning in adolescents. For example, we expected a significant influence of character maturity (especially of SD and of the combined score of SD and Co) on PWB as well as on different aspects of adjustment difficulties in the SDQ. Moreover, we also expect significant positive relations between temperament traits such as HA, which is positively correlated with neuroticism (*De Fruyt, Van De Wiele & Van Heeringen, 2000*), and psychosocial maladjustment.

## MATERIALS & METHODS

### Participants

Seventy two adolescents (63.9% of females) aged between 16 and 20 years (mean: 17.5 ± 0.75) were recruited in two Italian five-year high-school (a technical high school and a scientific and linguistic high school) (i.e., attending grades 3rd, 56.9% of the sample, and 4th) located in two small cities (approximately 7,000 and 14,000 inhabitants) in the north-east of Italy (Lignano e Latisana). All participants were volunteers previously informed about research purposes following schools procedures. All recruited participants reported no past history of neurological or mental illness and had no previous experience with the outcome measures used in the study. This study was approved by the Ethics Committee of the University of Udine (Ethical Application Ref: CGPER-2016-11-23-01) and all procedures performed in the study were in accordance with the ethical standards of the 1964 Helsinki declaration and its later amendments. Adults participants and parents of minor ones provided a written informed consent for the research assessment.

### Procedure

The questionnaires were administered in group in participants' classrooms, at the presence of their teacher and of a research psychologist. General instructions were provided in verbal form, while written instructions were reported above each questionnaire. Only minor compilation doubts were reported by participants and directly solved with minimal verbal information.

## Measures

Personality was assessed using the self-report form of the Temperament and Character Inventory in 125-items version (TCI-125; *Cloninger et al., 1994*), administered in Italian adaptation (*Delvecchio et al., 2016*). TCI-125 operationalizes Cloninger's personality model (*Cloninger, Svrakic & Przybeck, 1993*) with True/False items organized into four Temperament scales (i.e.: NS, Novelty Seeking; HA, Harm Avoidance; RD, Reward Dependence; Pe, Persistence) and three Character scales (i.e.: SD, Self-Directedness; Co, Cooperativeness; ST, Self-Transcendence).

SWB and PWB were measured respectively with the Italian adaptations of the Positive and Negative Affect Schedule (PANAS; *Watson, Clark & Tellegen, 1988*; *Terraciano, McCrae & Costa, 2003*) and of Psychological Well-Being scales (PWBs; *Ryff, 1989*; *Ruini et al., 2003a*). PANAS is composed by 20 5-point items organized into two scales of: Positive Affect, PA; and Negative Affect, NA. In this research, reference time was to the last week. PWBs comprises six 14-item scales with 6-point responses: Au, Autonomy; EM, Environmental Mastery; PG, Personal Growth; PR, Positive Relations; PL, Purpose in Life; SA, Self-Acceptance. A total scale (i.e., sum of all items) was also used (WB).

The self-report form for adolescent of the Strengths and Difficulties Questionnaire (SDQ-A; *Goodman, 1997*; *Goodman, 2001*) was used as a reliable measure covering the most important domains of psychopathology in normal school-age population (*Goodman, 2000*). The SDQ-A provides a Total difficulties (TOT) score, derived from four 5-item problem-scales scored on three levels and organized in two high order scales: INT, Internalized problems (sum of: Emotional symptoms, EMO, and Peer problems, PEE); EXT, Externalized problems (sum of: Conduct problems, BEH, and Hyperactivity/Inattention, HYP). In the SDQ-A, strengths are measured with a 5-item Prosocial behaviours (PRO) scale. Finally, the questionnaire also includes an optional sheet to measure: (i) subjective evaluation (four levels, from absent to severe); (ii) general impact (10-point scale); (iii) impact on others (four levels, from absent to main); and (iv) timing of onset (four levels, from last month to more than a year) of any reported difficulty.

## Data analysis

For all questionnaires, a scale score was not imputed if more than 10% of scale items were omitted (or more than one item in 5-item scales). Only one participant (female, aged 17 years) missed six TCI-125 scales. In other cases, scale-mean substitution was used to manage missing items (0.2% of all mandatory items). Pair-wise deletion was adopted in analyses.

Continuous measures were summarized reporting mean, standard deviation (SD), and range of variation of raw scores. Referring to score distribution from Italian adaptation (*Delvecchio et al., 2016*), an average level of TCI-125 scores was defined by the $\pm 0.440$ $z$-score interval (roughly corresponding to the central third of score distribution by sex and age; considering low scores those $\leq$33th percentile and high scores those $\geq$66th one; following *Cloninger, Svrakic & Przybeck, 1993*). A mature character was identified by having high scores in both SD and Co, while an immature character by low scores in the same two scales.

By-sex differences were tested with t-tests using Welch's correction for unequal variances. Linear models were fitted on well-being (i.e., both SWB and PWB) and problems scales (i.e., from SDQ-A, also considering PRO scale) using sex (dichotomized with 'Male' as '1') and age in years as covariates and TCI-125 scales as main predictors. Backward selection was adopted to select best final models. For final models, significance, adjusted coefficient of determination ($R^2$), and model coefficients (Bs; with their 95% confidence intervals, 95% c.i., significance, and corresponding standardized values, $\beta$s) were reported. Variance inflation factors (VIFs) were calculated to avoid multicollinearity in initial and final models (i.e., VIF was considered too high if $\geq 5$ and discussed if $\geq 2$).

Specific effects of character maturity (i.e., mature, average and immature character) on well-being and problems scales were tested with one-way univariate analyses of covariance (ANCOVAs), using sex and age in years as covariates. Partial omega-squared ($\omega_p^2$) was reported as effect-size measure, conventionally considering: 0.01, 0.06, and 0.14 as thresholds for small, medium, and large effect, respectively. Post-hoc analyses were carried out with Tukey's honest significant difference method.

A conventional level of significance was adopted ($\alpha = 0.05$), using Bonferroni's method to correct for multiple comparisons in regression models and in ANCOVAs (i.e., considering 20 independent scales, statistical significance was fixed to $p \leq 0.002$). Analyses were conducted using R 3.4.1 (*R Development Core Team, 2017*).

## RESULTS

Sample is described Table 1, together with personality assessment. Females showed statistically significant higher scores than males in HA ($t_{51.1} = 2.93$, $p = 0.005$) and RD ($t_{54.8} = 2.01$, $p = 0.049$). Emotional-behavioural problems (SDQ-A) and well-being (PWBs and PANAS) measures are reported in Table 2. Relative to males, females showed higher levels of EMO ($t_{60.4} = 4.25$, $p = 0.030$), INT ($t_{50.3} = 3.00$, $p = 0.004$), TOT ($t_{44.6} = 2.11$, $p = 0.040$), and NA ($t_{60.0} = 2.22$, $p = 0.030$). A mature character was reported by the 16.7% of the sample, while the 23.6% of it showed an immature character, without statistically significant differences by sex ($\chi_2^2 = 0.26$, $p = 0.879$), nor correlation between age and SD + Co score ($r = -0.063$, $p = 0.600$).

With regards to the optional SDQ-A sheet, no-problems were reported by 12.5% of participants, minor ones were reported by 66.7% of them, and moderate ones by 19.4% of adolescents. Only one participant (female, aged 18 years) indicated severe difficulties. General impact of reported difficulties scored between 0 and 9 ($2.43 \pm 2.108$), with moderate-to-main effects on others in the 14.3% of the cases. Moreover, 58.7% of these reported problems lasted for more than a year, 19.0% for more than six months, and 3.2% were recent problems (i.e., last month).

### Effects of temperament and character on well-being

All initial models testing for the effects of TCI-125 scales on PWBs and PANAS were statistically significant, with WB model showing highest fit ($R^2 = 0.737$; $F_{9,61} = 18.99$, $p < 0.001$), followed by PWB scales (Au: $R^2 = 0.707$; $F_{9,61} = 16.33$, $p < 0.001$; EM: $R^2 = 0.622$; $F_{9,61} = 11.14$, $p < 0.001$; SA: $R^2 = 0.598$; $F_{9,61} = 10.08$, $p < 0.001$; PL: $R^2 = $

**Table 1  Sample description with personality assessment (TCI-125).**

| | | | Mean ± SD [min; Max] |
|---|---|---|---|
| $N = 72$ (F: 46) | Age (years) | | All: 17.54 ± 0.749 [16; 20]<br>F: 17.41 ± 0.717 [16; 20]<br>M: 17.77 ± 0.765 [17; 20] |
| TCI-125 | *Temperament* | NS | All: 9.39 ± 3.503 [1; 17]<br>F: 9.22 ± 3.597 [1; 17]<br>M: 9.69 ± 3.38 [3; 16] |
| | | HA | All: 11.29 ± 5.124 [1; 20]<br>F: 12.58 ± 4.812[*] [3; 20]<br>M: 9.05 ± 4.952 [1; 20] |
| | | RD | All: 7.68 ± 2.848 [1; 14]<br>F: 8.18 ± 2.847[*] [1; 14]<br>M: 6.82 ± 2.687 [2; 12] |
| | | Pe | All: 2.71 ± 1.638 [0; 5]<br>F: 2.96 ± 1.623 [0; 5]<br>M: 2.3 ± 1.609 [0; 5] |
| | *Character* | SD | All: 14.42 ± 5.461 [2; 25]<br>F: 13.62 ± 5.417 [4; 25]<br>M: 15.8 ± 5.36 [2; 25] |
| | | Co | All: 17.26 ± 4.151 [5; 23]<br>F: 17.7 ± 3.363 [10; 23]<br>M: 16.5 ± 5.233 [5; 23] |
| | | ST | All: 6.55 ± 3.341 [0; 14]<br>F: 7.07 ± 3.454 [0; 14]<br>M: 5.65 ± 2.993 [0; 11] |

**Notes.**

TCI-125, Temperament and Character Inventory, self-report form, 125-items version; F, Female; M, Male; Max, Maximum observed value; min, Minimum observed value; NS, Novelty Seeking; HA, Harm Avoidance; RD, Reward Dependence; Pe, Persistence; SD, Self-Directedness; Co, Cooperativeness; ST, Self-Transcendence.

*Female scored higher than males (with $p < 0.05$).

0.571; $F_{9,61} = 9.01$, $p < 0.001$; PG: $R^2 = 0.501$; $F_{9,61} = 6.81$, $p < 0.001$; PR: $R^2 = 0.486$; $F_{9,61} = 6.40$, $p < 0.001$), and by SWB ones (NA: $R^2 = 0.421$; $F_{9,61} = 4.93$, $p < 0.001$; PA: $R^2 = 0.356$; $F_{9,61} = 3.74$, $p = 0.001$). No statistically significant effect was observed for sex covariates (all with $p > 0.050$), while age negatively predicted EM ($\beta = -0.219$; $B = -3.03$; $t_{61} = -2.62$, $p = 0.011$).

Final models are reported in Table 3. A high multicollinearity on HA was observed for all initial models (VIF = 2.29), and it was maintained in the final model for PG only (VIF = 2.28). Thus, predictive effect of HA in final PG model could be inflated by 56.2%.

In sum, specific temperamental and character traits, and in particular HA, SD and also ST have a widespread effect on PWB and SWB, with the same and other personality traits (e.g., NS, RD) helping to predict more specific facets of adolescents' well-being. Strikingly, SD was associated with all aspects of PWB and NA. Finally, to be female positively predicted Au.

ANCOVAs results on character maturity showed large statistically significant effects on most of PWBs scales (WB: $F_{2,67} = 14.23$, $p < 0.001$, $\omega_p^2 = 0.27$; EM: $F_{2,67} = 9.82$, $p < 0.001$, $\omega_p^2 = 0.20$; PG: $F_{2,67} = 9.91$, $p < 0.001$, $\omega_p^2 = 0.20$; PL: $F_{2,67} = 8.73$, $p < 0.001$, $\omega_p^2 = 0.18$; SA: $F_{2,67} = 9.90$, $p < 0.001$, $\omega_p^2 = 0.20$), with trends toward statistical significance in Au

**Table 2  Problems (SDQ-A) and well-being (PWBs and PANAS) assessment of the sample.**

| | | Mean ± SD [min; Max] | | | Mean ± SD [min; Max] |
|---|---|---|---|---|---|
| | | *Emotional-behavioural problems* | | | *Subjective and psychological well-being* |
| SDQ-A | TOT | All: 14.28 ± 6.147 [1; 28]<br>F: 15.46 ± 5.561* [3; 25]<br>M: 12.19 ± 6.675 [1; 28] | PWBs | WB | All: 341.28 ± 50.587 [233; 434]<br>F: 336.08 ± 52.086 [233; 434]<br>M: 350.46 ± 47.403 [243; 433] |
| | INT | All: 7.15 ± 3.931 [0; 18]<br>F: 8.15 ± 3.669* [1; 18]<br>M: 5.38 ± 3.817 [0; 16] | | Au | All: 59.01 ± 11.586 [34; 76]<br>F: 58.41 ± 12.185 [34; 76]<br>M: 60.08 ± 10.590 [35; 74] |
| | EXT | All: 7.12 ± 3.335 [0; 17]<br>F: 7.3 ± 3.076 [1; 12]<br>M: 6.81 ± 3.795 [0; 17] | | EM | All: 53.59 ± 10.339 [27; 74]<br>F: 52.61 ± 10.678 [27; 70]<br>M: 55.33 ± 9.664 [33; 74] |
| | EMO | All: 4.26 ± 2.742 [0; 10]<br>F: 5.15 ± 2.641* [0; 10]<br>M: 2.69 ± 2.187 [0; 10] | | PG | All: 62.24 ± 8.707 [36; 82]<br>F: 62.48 ± 8.709 [48; 77]<br>M: 61.81 ± 8.859 [36; 82] |
| | PEE | All: 2.89 ± 2.046 [0; 8]<br>F: 3.00 ± 2.011 [0; 8]<br>M: 2.69 ± 2.131 [0; 8] | | PR | All: 58.05 ± 12.008 [26; 83]<br>F: 56.78 ± 12.198 [26; 80]<br>M: 60.29 ± 11.555 [33; 83] |
| | BEH | All: 2.9 ± 2.022 [0; 10]<br>F: 3.09 ± 1.799 [0; 6]<br>M: 2.58 ± 2.369 [0; 10] | | PL | All: 57.14 ± 10.395 [33; 76]<br>F: 56.38 ± 10.939 [33; 76]<br>M: 58.50 ± 9.408 [40; 76] |
| | HYP | All: 4.22 ± 1.848 [0; 10]<br>F: 4.22 ± 1.825 [1; 7]<br>M: 4.23 ± 1.925 [0; 10] | | SA | All: 51.24 ± 13.32 [24; 80]<br>F: 49.41 ± 13.973 [24; 80]<br>M: 54.47 ± 11.637 [24; 79] |
| | PRO | All: 7.25 ± 1.998 [2; 10]<br>F: 7.07 ± 2.070 [2; 10]<br>M: 7.58 ± 1.858 [4; 10] | PANAS | PA | All: 29.47 ± 6.836 [14; 44]<br>F: 28.89 ± 7.230 [14; 44]<br>M: 30.50 ± 6.075 [17; 40] |
| | | | | NA | All: 20.51 ± 7.463 [10; 45]<br>F: 21.87 ± 7.713* [11; 45]<br>M: 18.10 ± 6.452 [10; 36] |

**Notes.**
SDQ-A, Strengths and Difficulties Questionnaire for Adolescents, self-completion form; PWBs, Psychological Well-Being scales; PANAS, Positive and Negative Affect Schedule; TOT, Total difficulties; INT, Internalising problems; EXT, Externalising problems; EMO, Emotional symptoms; PEE, Peer problems; BEH, Conduct problems; HYP, Hyperactivity/Inattention; PRO, Prosocial behaviours; WB, Total well-being; Au, Autonomy; EM, Environmental Mastery; PG, Personal Growth; PR, Positive Relations; PL, Purpose in Life; SA, Self-Acceptance; PA, Positive Affect; NA, Negative Affect; F, Female; M, Male; Max, Maximum observed value; min, Minimum observed value.
*Female scored higher than males (with $p < 0.05$).

$(F_{2,67} = 4.78, p = 0.012, \omega_p^2 = 0.09)$, PA $(F_{2,67} = 3.16, p = 0.049, \omega_p^2 = 0.06)$, and NA $(F_{2,67} = 5.08, p = 0.009, \omega_p^2 = 0.10)$. No statistically significant effect was observed for PR $(F_{2,67} = 3.00, p = 0.057, \omega_p^2 = 0.05)$. Post-hoc results are summarized in Fig. 1. Only a trend toward statistical significance resulted for sex in NA analysis $(F_{1,67} = 4.89, p = 0.030, \omega_p^2 = 0.05)$, with higher scores in females.

## Effects of temperament and character on psychosocial adjustment

Initial models testing for the effects of TCI-125 scales on SDQ-A were statistically significant for summarizing scales (TOT: $R^2 = 0.499$; $F_{9,61} = 6.74$, $p < 0.001$; INT: $R^2 = 0.530$; $F_{9,61} = 7.65$, $p < 0.001$; EXT: $R^2 = 0.414$; $F_{9,61} = 4.8$, $p < 0.001$), for EMO $(R^2 = 0.600$; $F_{9,61} = 10.18$, $p < 0.001)$, BEH $(R^2 = 0.336$; $F_{9,61} = 3.43$, $p = 0.002)$, and PRO $(R^2 = 0.329$; $F_{9,61} = 3.32$, $p = 0.002)$. Models for PEE and HYP trend to statistical significance
**Table 3  Prediction of PWBs and PANAS scores by TCI-125 scales (final linear models after backward selection of predictors). Sex and age in years were introduced as covariates in all initial models.**

| Predicted model: $R^2$; $F_{df}$, $p$ | Predictor | $\beta$ | B (±95% c.i.) | $t_{df}$, $p$ |
|---|---|---|---|---|
| **WB** | HA | −0.181 | −1.79 (−3.32, −0.26) | $t_{66} = -2.33, p = 0.023$* |
| $R^2 = 0.731$ | SD | +0.734 | +6.80 (+5.37, +8.23) | $t_{66} = 9.49, p < 0.001$* |
| $F_{4,66} = 44.73, p < 0.001$ | ST | +0.187 | +2.83 (+0.87, +4.79) | $t_{66} = 2.88, p = 0.005$* |
|  | Age (years) | −0.128 | −8.61 (−17.36, +0.15) | $t_{66} = -1.96, p = 0.054$ |
| **Au** | NS | +0.217 | +0.71 (+0.26, +1.16) | $t_{66} = 3.16, p = 0.002$* |
| $R^2 = 0.698$ | RD | −0.542 | −2.19 (−2.76, −1.62) | $t_{66} = -7.71, p < 0.001$* |
| $F_{4,66} = 38.13, p < 0.001$ | SD | +0.630 | +1.32 (+1.04, +1.61) | $t_{66} = 9.13, p < 0.001$* |
|  | Sex ('Male' = 1) | −0.177 | −4.18 (−7.54, −0.82) | $t_{66} = -2.48, p = 0.016$* |
| **EM** | NS | −0.175 | −0.52 (−1.02, −0.02) | $t_{66} = -2.07, p = 0.042$* |
| $R^2 = 0.604$ | HA | −0.337 | −0.69 (−1.1, −0.27) | $t_{66} = -3.29, p = 0.002$* |
| $F_{4,66} = 25.21, p < 0.001$ | SD | +0.535 | +1.02 (+0.66, +1.38) | $t_{66} = 5.60, p < 0.001$* |
|  | Age (years) | −0.245 | −3.39 (−5.55, −1.23) | $t_{66} = -3.13, p = 0.003$* |
| **PG** | NS | +0.153 | +0.38 (−0.13, +0.89) | $t_{63} = 1.48, p = 0.144$ |
| $R^2 = 0.501$ | HA | −0.200 | −0.34 (−0.8, +0.12) | $t_{63} = -1.48, p = 0.143$ |
| $F_{7,63} = 9.03, p < 0.001$ | RD | −0.191 | −0.59 (−1.25, +0.08) | $t_{63} = -1.75, p = 0.084$ |
|  | SD | +0.461 | +0.74 (+0.35, +1.12) | $t_{63} = 3.83, p < 0.001$* |
|  | Co | +0.156 | +0.33 (−0.12, +0.78) | $t_{63} = 1.46, p = 0.149$ |
|  | ST | +0.214 | +0.56 (+0.05, +1.07) | $t_{63} = 2.21, p = 0.031$* |
|  | Sex ('Male' = 1) | −0.176 | −3.17 (−6.67, +0.34) | $t_{63} = -1.81, p = 0.076$ |
| **PR** | NS | +0.164 | +0.56 (−0.06, +1.18) | $t_{66} = 1.80, p = 0.076$ |
| $R^2 = 0.468$ | RD | +0.454 | +1.92 (+1.13, +2.71) | $t_{66} = 4.86, p < 0.001$* |
| $F_{4,66} = 14.49, p < 0.001$ | SD | +0.465 | +1.03 (+0.62, +1.43) | $t_{66} = 5.08, p < 0.001$* |
|  | Sex ('Male' = 1) | +0.154 | +3.82 (−0.86, +8.50) | $t_{66} = 1.63, p = 0.108$ |
| **PL** | NS | −0.170 | −0.50 (−1.00, −0.01) | $t_{66} = -2.05, p = 0.045$* |
| $R^2 = 0.547$ | SD | +0.651 | +1.25 (+0.93, +1.57) | $t_{66} = 7.81, p < 0.001$* |
| $F_{4,66} = 19.94, p < 0.001$ | ST | +0.335 | +1.05 (+0.53, +1.57) | $t_{66} = 4.00, p < 0.001$* |
|  | Age (years) | −0.153 | −2.12 (−4.45, +0.20) | $t_{66} = -1.82, p = 0.073$ |
| **SA** | HA | −0.244 | −0.63 (−1.13, −0.13) | $t_{67} = -2.53, p = 0.014$* |
| $R^2 = 0.571$ | SD | +0.581 | +1.42 (+0.94, +1.89) | $t_{67} = 6.00, p < 0.001$* |
| $F_{3,67} = 29.7, p < 0.001$ | ST | +0.193 | +0.77 (+0.13, +1.41) | $t_{67} = 2.40, p = 0.019$* |
| **PA** | NS | −0.192 | −0.38 (−0.81, +0.06) | $t_{67} = -1.74, p = 0.087$ |
| $R^2 = 0.336$ | HA | −0.571 | −0.77 (−1.05, −0.48) | $t_{67} = -5.36, p < 0.001$* |
| $F_{3,67} = 11.28, p < 0.001$ | Pe | +0.186 | +0.78 (−0.09, +1.65) | $t_{67} = 1.79, p = 0.079$ |
| **NA** | HA | +0.212 | +0.31 (−0.03, +0.65) | $t_{66} = 1.83, p = 0.072$ |
| $R^2 = 0.399$ | Pe | +0.207 | +0.95 (+0.05, +1.84) | $t_{66} = 2.10, p = 0.040$* |
| $F_{4,66} = 10.94, p < 0.001$ | SD | −0.438 | −0.60 (−0.92, −0.28) | $t_{66} = -3.72, p < 0.001$* |
|  | ST | +0.137 | +0.31 (−0.13, +0.74) | $t_{66} = 1.41, p = 0.163$ |

**Notes.**

TCI-125, Temperament and Character Inventory, self-report form, 125-items version; PWBs, Psychological Well-Being scales; PANAS, Positive and Negative Affect Schedule; WB, Total well-being; Au, Autonomy; EM, Environmental Mastery; PG, Personal Growth; PR, Positive Relations; PL, Purpose in Life; SA, Self-Acceptance; PA, Positive Affect; NA, Negative Affect; NS, Novelty Seeking; HA, Harm Avoidance; RD, Reward Dependence; Pe, Persistence; SD, Self-Directedness; Co, Cooperativeness; ST, Self-Transcendence.

*Statistically significant B of the predictor.

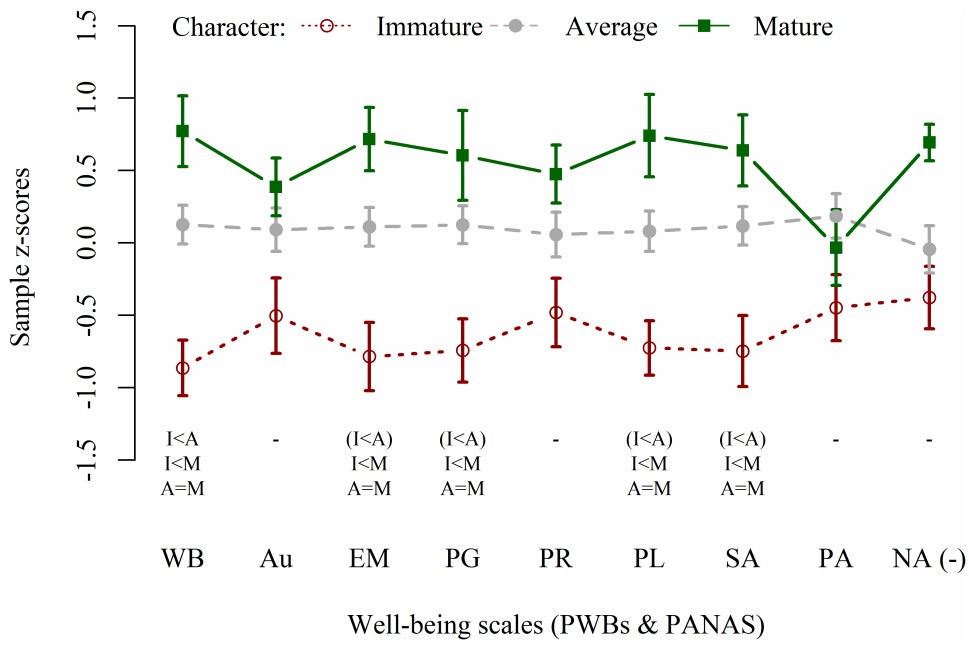

**Figure 1** **Well-being profiles of participants with mature, average, and immature character.** PWBs, Psychological Well-Being scales; PANAS, Positive and Negative Affect Schedule; WB, Total well-being; Au, Autonomy; EM, Environmental Mastery; PG, Personal Growth; PR, Positive Relations; PL, Purpose in Life; SA, Self-Acceptance; PA, Positive Affect; NA (−), Negative Affect (scale score multiplied by −1). I/A/M, Immature, Average, and Mature character in post-hoc comparisons; trends to statistical significance (i.e., $p$ ranging 0.050–0.002) are written between-braces.

(respectively: $R^2 = 0.305$; $F_{9,61} = 2.98$, $p = 0.005$; and $R^2 = 0.316$; $F_{9,61} = 3.14$, $p = 0.004$). To be female predicted INT ($\beta = 0.213$; $B = 1.73$; $t_{61} = 2.13$, $p = 0.037$) and EMO ($\beta = 0.219$; $B = 1.25$; $t_{61} = 2.38$, $p = 0.020$), while no statistically significant age effect was observed. Final models are reported in Table 4. High multicollinearity on HA disappeared in final models.

In sum, SD protects against all considered poor adjustment features and difficulties, excluding pro-social behaviours. Also, cooperativeness is associated with psychosocial adjustment. In particular, it has significant positive associations with pro-social behaviours and marginally significant negative associations with total difficulties and externalizing problems. Moreover, different emotional-behavioural problems are associated with specific temperamental traits such as NS, HA, and RD. Finally, to be female positively predicted INT and EMO.

ANCOVAs results on character maturity showed large statistically significant effects on total SDQ-A scales (TOT: $F_{2,67} = 16.77$, $p < 0.001$, $\omega_p^2 = 0.30$; INT: $F_{2,67} = 10.35$, $p < 0.001$, $\omega_p^2 = 0.21$; EXT: $F_{2,67} = 10.67$, $p < 0.001$, $\omega_p^2 = 0.21$), EMO ($F_{2,67} = 9.77$, $p < 0.001$, $\omega_p^2 = 0.20$), and BEH ($F_{2,67} = 8.94$, $p < 0.001$, $\omega_p^2 = 0.18$). Trends to statistical significance were observed in the other scales (HYP: $F_{2,67} = 5.97$, $p = 0.004$, $\omega_p^2 = 0.12$; PEE: $F_{2,67} = 3.56$, $p < 0.034$, $\omega_p^2 = 0.07$; PRO: $F_{2,67} = 4.21$, $p < 0.019$, $\omega_p^2 = 0.08$). Problems profiles and post-hoc results are reported in Fig. 2. A statistical significant effect of sex,

**Table 4  Prediction of SDQ-A scores by TCI-125 scales (final linear models after backward selection of predictors).** Sex and age in years were introduced as covariates in all initial models.

| Predicted model: $R^2$; $F_{df}$, $p$ | Predictor | $\beta$ | B (±95% ci) | $t_{df}$, $p$ |
|---|---|---|---|---|
| **TOT** | SD | −0.583 | −0.66 (−0.87, −0.45) | $t_{67} = -6.19, p < 0.001^*$ |
| $R^2 = 0.476$ | Co | −0.163 | −0.24 (−0.52, +0.03) | $t_{67} = -1.75, p = 0.085$ |
| $F_{3,67} = 20.31, p < 0.001$ | Sex ('Male' = 1) | −0.174 | −2.22 (−4.55, +0.12) | $t_{67} = -1.89, p = 0.063$ |
| **INT** | NS | −0.141 | −0.16 (−0.37, +0.05) | $t_{65} = -1.50, p = 0.138$ |
| $R^2 = 0.525$ | HA | +0.239 | +0.18 (0.00, +0.37) | $t_{65} = 2.00, p = 0.050^*$ |
| $F_{5,65} = 14.35, p < 0.001$ | SD | −0.414 | −0.30 (−0.46, −0.14) | $t_{65} = -3.68, p < 0.001^*$ |
| | Co | −0.134 | −0.13 (−0.3, +0.05) | $t_{65} = -1.46, p = 0.150$ |
| | Sex ('Male' = 1) | −0.201 | −1.63 (−3.12, −0.14) | $t_{65} = -2.19, p = 0.032^*$ |
| **EXT** | NS | +0.375 | +0.36 (+0.18, +0.54) | $t_{67} = 3.97, p < 0.001^*$ |
| $R^2 = 0.407$ | SD | −0.446 | −0.27 (−0.39, −0.15) | $t_{67} = -4.58, p < 0.001^*$ |
| $F_{3,67} = 15.35, p < 0.001$ | Co | −0.166 | −0.13 (−0.29, +0.02) | $t_{67} = -1.70, p = 0.095$ |
| **EMO** | HA | +0.397 | +0.21 (+0.11, +0.32) | $t_{67} = 4.06, p < 0.001^*$ |
| $R^2 = 0.591$ | SD | −0.355 | −0.18 (−0.27, −0.08) | $t_{67} = -3.77, p < 0.001^*$ |
| $F_{3,67} = 32.30, p < 0.001$ | Sex ('Male' = 1) | −0.235 | −1.34 (−2.28, −0.40) | $t_{67} = -2.83, p = 0.006^*$ |
| **PEE** | NS | −0.241 | −0.14 (−0.26, −0.02) | $t_{67} = -2.28, p = 0.026^*$ |
| $R^2 = 0.263$ | RD | −0.238 | −0.17 (−0.32, −0.02) | $t_{67} = -2.24, p = 0.028^*$ |
| $F_{3,67} = 7.95, p < 0.001^{**}$ | SD | −0.374 | −0.14 (−0.22, −0.06) | $t_{67} = -3.56, p = 0.001^*$ |
| **BEH** | NS | +0.334 | +0.19 (+0.08, +0.31) | $t_{68} = 3.30, p = 0.002^*$ |
| $R^2 = 0.300$ | SD | −0.443 | −0.16 (−0.24, −0.09) | $t_{68} = -4.36, p < 0.001^*$ |
| $F_{2,68} = 14.60, p < 0.001$ | | | | |
| **HYP** | NS | +0.342 | +0.18 (+0.07, +0.29) | $t_{68} = 3.30, p = 0.002^*$ |
| $R^2 = 0.269$ | SD | −0.399 | −0.14 (−0.21, −0.07) | $t_{68} = -3.85, p < 0.001^*$ |
| $F_{2,68} = 12.53, p < 0.001^{**}$ | | | | |
| **PRO** | HA | −0.420 | −0.16 (−0.25, −0.08) | $t_{67} = -3.88, p < 0.001^*$ |
| $R^2 = 0.291$ | RD | +0.314 | +0.22 (+0.06, +0.39) | $t_{67} = 2.68, p = 0.009^*$ |
| $F_{3,67} = 9.16, p < 0.001$ | Co | +0.231 | +0.11 (0.00, +0.22) | $t_{67} = 2.06, p = 0.043^*$ |

Notes.

TCI-125, Temperament and Character Inventory, self-report form, 125-items version; SDQA, Strengths and Difficulties Questionnaire for Adolescents, self-completion form; TOT, Total difficulties; INT, Internalising problems; EXT, Externalising problems; EMO, Emotional symptoms; PEE, Peer problems; BEH, Conduct problems; HYP, Hyperactivity/Inattention; PRO, Prosocial behaviours; NS, Novelty Seeking; HA, Harm Avoidance; RD, Reward Dependence; Pe, Persistence; SD, Self-Directedness; Co, Cooperativeness; ST, Self-Transcendence.

*Statistically significant B of the predictor.

**Initial model (i.e., before backward selection) was not statistically significant (Bonferroni correction applied).

with females scoring higher than males, was found for INT ($F_{1,67} = 10.94$, $p = 0.002$, $\omega_p^2 = 0.13$ medium) and EMO ($F_{1,67} = 17.15$, $p < 0.001$, $\omega_p^2 = 0.21$ large), and a trend toward statistical significance was found for TOT ($F_{1,67} = 6.65$, $p = 0.012$, $\omega_p^2 = 0.08$). Participants' age showed a trend toward statistical significance in EMO model only ($F_{1,67} = 4.01$, $p = 0.049$, $\omega_p^2 = 0.01$), with lower scores in older participants.

## DISCUSSION

The aim of this study was to investigate the question of temperament and character's relation to well-being and psychosocial adjustment in adolescence. This was done by asking a sample of Italian late adolescents to self-report their personality (TCI-125;
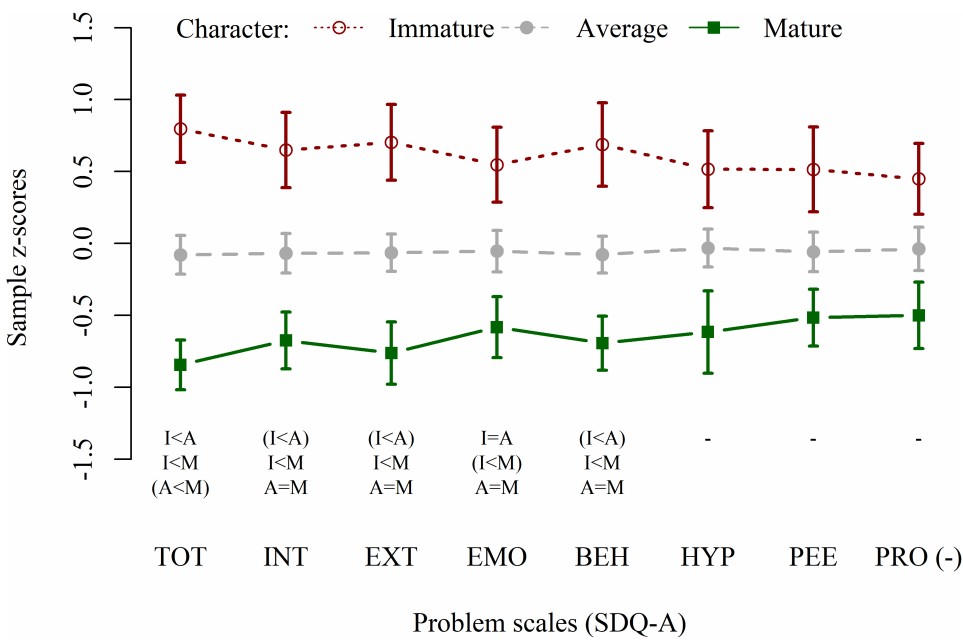

**Figure 2** **Emotional-behavioural problems profiles of participants with mature, average, and immature character.** SDQ-A, Strengths and Difficulties Questionnaire for Adolescents, self-completion form; TOT, Total difficulties; INT, Internalising problems; EXT, Externalising problems; EMO, Emotional symptoms; PEE, Peer problems; BEH, Conduct problems; HYP, Hyperactivity/Inattention; PRO (−), Prosocial behaviours (scale score multiplied by −1). I/A/M, Immature, Average, and Mature character in post-hoc comparisons; trends to statistical significance (i.e., $p$ ranging 0.050–0.002) are written between-braces.

Cloninger, Svrakic & Przybeck, 1993), psychological well-being (PWBs; *Ryff, 1989*), subjective well-being (SWB; *Watson, Clark & Tellegen, 1988*), and psychosocial adjustment (SDQ-A; *Goodman, 1997*; *Goodman, 2001*). Prediction of PWBs, SWB (PANAS), and SDQ-A scores by TCI-125 scales showed that temperament and character dimensions significantly predict different aspects of well-being and psychosocial functioning in adolescents.

Concerning adolescents' character, the present findings highlight the usefulness of including indexes of its maturity, at the intrapersonal, interpersonal and transpersonal levels, in the study of adolescents' well-being and psychosocial functioning, possibly considering also their combination (SD and Co). Previous recent studies have been focused in particular on the link between character and SWB in adolescence (*Gillham et al., 2011*; *Garcia & Moradi, 2012*; *Garcia, Kerekes & Archer, 2012*; *Nima, Archer & Garcia, 2012*; *Schültz, Archer & Garcia, 2013*; but see *Garcia, 2011*; *Moreira et al., 2015*), or, separately, on the relation between character strength and emotional-behavioural problems (*Bromley, Johnson & Cohen, 2006*; *Gillham et al., 2011*; *Garcia et al., 2012*; see 'Introduction'). In the present study, the assessment of SWB and different facets of PWB, as well as the measurement of adolescents' strength and emotional-behavioural difficulties, has allowed to reveal specific associations between different aspects of character and adolescents' well-being and psychosocial adjustment. Thus, we found that Self-Directedness (SD) had

a widespread protective effect on PWB and emotional-behavioural problems (Tables 3 and 4). It also negatively predicted negative affect (NA in the PANAS). Extending previous similar findings (*Cloninger, 2006*; *Garcia, 2011*; *Kerekes et al., 2017*), the current data suggest that SD may allow adolescents to strengthen, to cope better with difficult situations such as peer pressure, and to behave in line with their long-term goals and values, probably because of good abilities and feelings of self-discipline, self-acceptance, effectiveness, and self-esteem. More generally, these findings may underline the importance of interventions in adolescence that, delivered at home, at schools or in clinical settings, aim at promoting and strengthening self-acceptance, as well as a sense of mastery and hope for self-directed behaviours and attitudes (*Cloninger, 2006*; *Garcia, 2011*).

In addition to SD, we found that the transpersonal and interpersonal levels of self-maturity, measured respectively by the Self-Transcendence (ST) and Cooperativeness (Co) character scales of the TCI, were associated with specific aspects of adolescents' well-being and strengths. Tellingly, PWBs constructs associated to a sense of continued growth and realization (Personal growth, PG), to the belief that one's life is meaningful and purposeful (Purpose in life, PL), and to the positive evaluation of oneself and one's past life (Self-acceptance; SA), were positively influenced by holistic and transcendental beliefs (ST) (Table 3). This in turn suggests the importance to foster spiritual development in adolescents while they move along their path to well-being (*Cloninger, 2006*). Notwithstanding, we found that ST was not predictive of adolescents' psychosocial adjustment (Table 4); moreover, although not significant, there was a marginal positive relation between ST and NA. In addition to provide partial support to previously reported associations between ST and psychiatry problems in adolescents (*Consoli et al., 2015*) and schizotypy traits in adults (*Brambilla et al., 2014*), these data indicate that the complex relation between ST, well-being and psychopathology is in need of further investigation (see also *Cloninger, 2006*; *Cloninger & Zohar, 2011*).

From a complimentary perspective, maturation of the self in aspects related to empathy, kindness, and forgiveness (Co in the TCI) was positively associated with pro-social behaviours (PRO) and more marginally and negatively with total difficulties and externalizing problems (Table 4). Similarly, RD temperament (a TCI trait reflecting the tendency to respond markedly to signals of reward, particularly to signals of social approval, support, and sentiment) increased pro-social behaviours and reduced relational problems with peers (respectively PRO and PEE in the SDQ-A). Also, we found that RD was positively associated with relational aspects of PWB (i.e., the PR scale measuring the possession of quality relations with others such as warm and trusting) and negatively with a sense of self-determination (Autonomy, Au in the PWBs) (Tables 3 and 4). Overall, these data indicate that another key element to understand well-being and psychosocial adjustment in adolescence seems to involve the possibility to give and receive help and to experience satisfying interactions with other individuals (*Garcia, 2011*).

Finally, comparing profiles of mature and immature character (i.e., combination of SD and Co levels), we observed that immature character was strongly associated with low levels of psychological well-being (on total WB, EM, PG, PL, and SA scales; see Fig. 1) and with increased risk for total, internalized and externalized problems (in particular on EMO and

BEH scales; see Fig. 2). This confirms the importance of considering maturity as protective factor for general psychological adjustment (*Svrakic et al., 1993*), even before and beyond the possible onset of chronic adult disorders.

Turning to other aspects of adolescents' temperament, we found that also Novelty Seeking (NS) and Harm Avoidance (HA) had a role in adolescents' social adjustment, with a reduction of relational problems with peers with increasing NS and of active pro-social behaviours with higher HA. Nevertheless, NS and HA also disclosed other interesting associations with adolescents' well-being and psychosocial functioning. Considering PWB, NS was positively associated with Au and negatively with PL and Environmental Mastery (EM) (Table 3). Overall, these results indicate that the tendency to explorative, enthusiastic, but even disorderly and impulsive behaviours (NS) in adolescence, may increase a sense of self-determination and independence but it can also lead to a reduced capacity to manage effectively surrounding world and one's life, which may lose meaning and a sense of directedness. Accordingly, it has been suggested that high novelty seeking scores in the TCI might be counterproductive in adolescence (e.g., *Garcia & Moradi, 2012*). With regard to psychosocial functioning, we found that NS was positively associated with externalized problems (i.e., Conduct problems, BEH, and Hyperactivity/Inattention, HYP in the SDQ-A) (Table 4), a result in line with recent past research showing higher scores on NS in children and adolescents with attention-deficit hyperactivity disorder (ADHD) and disruptive/aggressive behaviour disorders compared with healthy and clinical controls (*Drechsler et al., 2015*; *Gomez et al., 2017*; *Kerekes et al., 2017*).

From a complimentary perspective, temperament tendency to avoid unpleasant situations and being fearful, doubtful, pessimist, and worried (HA) played a significant role in reducing PWB and SWB (Table 3). HA negatively predicted positive affect (PA in the PANAS) and total PWB, similar to previous findings (*Garcia, 2011*). The analysis of the PWBs subscales showed negative associations between HA and EM and SA (Table 3). Moreover, concerning adolescents' psychosocial adjustment, HA positively predicted internalized problems (INT), and in particular Emotional symptoms (EMO) (Table 4). Noteworthy, recent findings on young adults (mean age: 24.49 years) have shown that HA is related to depression and anxiety and that such relation is partially mediated by certain types of dysfunctional meta-cognitive beliefs, such as negative beliefs about the uncontrollability of thoughts and danger (e.g., ''if I don't control my worries they will control me''; *Gawęda & Kokoszka, 2014*, p. 1036). Future studies may try to test whether similar dysfunctional meta-cognitive beliefs might become vulnerability factors of emotional distress and reduced senses of self-acceptance and control over external world (i.e., EM in the PWBs) also in adolescents, as possibly suggested by the present data.

The present study has a number of limitations that should be borne in mind when interpreting the findings. The first limitation concerns the restricted sample size, which although being similar to that of other previous related studies (*Garcia, 2011*; *Garcia & Moradi, 2012*), and somehow well distributed in terms of gender and attending grades of adolescents coming from two schools located in two different cities, suggests extension and replication of current results to larger samples of adolescents. Moreover, the sample size and the conservative correction for multiple comparison adopted allowed to detect

with sufficient statistical power only large-size effects. Another limitation concerns the assessment of personality, which was carried out using a short instrument (TCI-125), without the opportunity to evaluate facets of main personality dimensions (i.e., the subscales involved in the TCI 240-item version). This was done to avoid excessive burden in participants. However, TCI-125 has already been used in Italy, and with a similar school-aged sample, in a study showing the good reliability of this personality tool (*Delvecchio et al., 2016*). Moreover, in the assessment of the level of maturity of character, TCI-125 has already been adopted (*Kluger et al., 1999*).

A further connected issue pertains to the exclusive use of self-report measures made in the current study. It is in fact well known that these measures can be susceptible to desirable responding and are sometime considered to be less reliable than more objective measures, such as observation, for instance in detecting externalizing behaviours (*Hinshaw et al., 1992*; *Schwarz, 1999*). Despite other studies showing that behavioural problems can reliably be assessed using self-report (*Bartels et al., 2011*), it is advisable that future studies will extend the present findings by relying on multiple informants (adolescents, parents, and teachers) to assess externalizing behaviours, and by including, in addition to adolescents' self-reports, implicit measures of personality (which are more difficult to control or to fake; e.g., the Implicit Association Test, see *Crescentini et al., 2014*).

## CONCLUSIONS

In conclusion, despite its limitations, the current study shows the importance of continuing to assess personality's relation to well-being and psychosocial functioning in adolescence. We found that temperament and character dimensions are significantly associated with different aspects of well-being and psychosocial adjustment in adolescents. In particular, Self-Directedness, a crucial aspect of character maturity, had a widespread protective effect on well-being and emotional-behavioural problems. Combining Self-Directedness and Cooperativeness, we also showed a marked association between immaturity of character and low psychological well-being and psychosocial adjustment. More generally, the present results suggest the usefulness of continuing to evaluate (TCI) character dimensions in investigations focused on adolescents' well-being and psychosocial functioning, especially in the contexts of potential interventions aimed at enhancing development of adolescents' character dimensions at the intrapersonal, interpersonal, and transpersonal levels.

## ACKNOWLEDGEMENTS

The authors would like to acknowledge the generosity of the school ISIS ''E Mattei'' which allowed us to evaluate the students.

### Funding

The first and last authors were funded by the University of Udine. The second author was funded by the scientific institute IRCCS ''Eugenio Medea''. The third author was funded

by the ISIS ''Enrico Mattei''. The fourth author was funding by the University of Milan and partly supported by grants from the Italian Ministry of Health (RF-2011-02352308). There was no additional external funding received for this study. The funders had no role in study design, data collection and analysis, decision to publish, or preparation of the manuscript.

### Grant Disclosures

The following grant information was disclosed by the authors:
University of Udine.
Scientific institute IRCCS ''Eugenio Medea''.
ISIS ''Enrico Mattei''.
University of Milan.
Italian Ministry of Health: RF-2011-02352308.

### Competing Interests

Paolo Brambilla is an Academic Editor for PeerJ.

### Author Contributions

- Cristiano Crescentini conceived and designed the experiments, performed the experiments, analyzed the data, prepared figures and/or tables, authored or reviewed drafts of the paper, approved the final draft.
- Marco Garzitto conceived and designed the experiments, analyzed the data, prepared figures and/or tables, authored or reviewed drafts of the paper, approved the final draft.
- Andrea Paschetto performed the experiments, authored or reviewed drafts of the paper, approved the final draft.
- Paolo Brambilla and Franco Fabbro conceived and designed the experiments, authored or reviewed drafts of the paper, approved the final draft.

### Human Ethics

The following information was supplied relating to ethical approvals (i.e., approving body and any reference numbers):

This study was approved by the Ethics Committee of the University of Udine (Ethical Application Ref: CGPER-2016-11-23-01).

### Data Availability

Raw data is available in the Supplemental Information.

### Supplemental Information

Supplemental information for this article can be found online at http://dx.doi.org/10.7717/peerj.4484#supplemental-information.

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
