# Peer review of "Temperament and character effects on late adolescents’ well-being and emotional-behavioural difficulties"

_PeerJ, doi:10.7717/peerj.4484_

## Round 0.1 · original submission · Minor Revisions

· Academic Editor

Minor Revisions

Dear Cristiano,

This is a very interesting paper

As you can see Reviewer 1 basically thought the paper was fine as it is and Reviewer 2 felt there were corrections to be made to the presentation in the introduction and to the arguments in the Discussion.

Please address the reviewer s comments and resubmit.

Ada

·

Basic reporting

The writing is good except for a few typos and awkward phrases. Specifically,
page 7 of 37 in the pdf, line 52 should be "an underlying", not "un underlying". Line 55, trait, not traits. Line 69, no comma is needed. On page 8 of 37, line 88, refer to propositional or insight learning, not prepositional learning. On page 9 of 37, line 103, the word differentiated is unclear; perhaps you mean correlated? On page 10 of 37, line 134, "prevents from" is awkward, perhaps yo could change this to "makes it difficult to ascertain in the same subjects..."

The literature review is excellent and the article is well organized.

Experimental design

The study is cross-sectional with self-report on multiple instruments. The limitations of this are adequately discussed. Methods are clearly described. The authors rely on linear regression with individual scale scores. Future work could be enhanced by considering multidimensional profiles of traits, but what was done is adequate for the limited sample size.

Validity of the findings

The results are reasonable and extend prior findings. Conclusions are justified and clearly stated except that the authors consistently minimize the associations they observed for Cooperativeness. Specifically, CO was correlated with Personal Growth (PG) so it is false to say that Co was not associated with well-being on page 15 of 37, line 246. Likewise, on page 16 of 37, authors should add a statement that CO had significant associations with adjustment, including Total, externalizing, and Prosocial behaviors. Therefore in the discussion the paragraph about Cooperativeness needs to be revised to indicate that Co had effects primarily related to personal growth and prosocial behaviors. It is possible that a composite measure of (SD+CO) would be a stronger predictor than either alone because this is a validated indicator of character maturity and absence of personality disorder.

Additional comments

This is a well-done cross-sectional study in an important area, the well-being of adolescents. There are only one issue about interpretation about Cooperativeness that needs revision and some minor awkward language and typos that can be easily corrected. I hope you continue this line of investigation and encourage you to consider profiles of traits as well as individual traits.

Reviewer 2 ·

Basic reporting

The comments are included in the "general" section

Experimental design

Sample
More information on the sample (background variables) would be needed.
Measures
Speaking about “behavior” should be done with a caution because just self-reports have been utilized. After having read the “measures” a reader is still more convinced of an inaccurary between the content of an introduction and an implementation.
Data analysis is very traditional and would benefit from more advanced approach. A low number of subjects is, of course, an evident disadvantage

Validity of the findings

The findings per se are in no way novel but available to be derived from the previous literature. Therefore, some new approach or advanced statistical model would be needed.

Additional comments

Temperament and character effects on late adolescents’ well-being and emotional-behavioral difficulties

This study examines associations between personality and psychological and subjective well-being. Several aspects of the study need reconsideration.
Introduction
“Goodness of fit” between the introduction and the study problem is not the best one, i.e. the introduction doesn’t lead to this very research problem in the best way but describes findings which do not help the reader to get the focus of this study. Actually there are references to previous literature which are more likely to mislead the reader than justify the present study (or a refining the study problem is needed). The most relevant literature in this context has been in many respects passed over. A distinction between psychological well-being and subjective well-being is factitious and therefore, not clear. Actually, the overlapping is high resulting in strange claims on a role of personality in PWE and SWE (i.e. claims on a different role of a personality in PWE from a role in SWE). Other concept used in the present study should be clarified, too, for instance a lack of symptoms is not identical with well-being and well-being with psychosocial adjustment. The concepts to be used are far from clear. – More specified hypotheses are possible to be derived from the previous literature.
.

Discussion
There are major concerns in discussion. At least the following revisions would lead to remarkable improvements. Using the concepts should be consistent. Extroversion shares elements with novelty seeking, but however, novelty seeking cannot be in midstream called extroverted behavior. Inconsistency is true with most of the concepts. It is not always easy to see, whether the authors speak about the findings of the present study or whether they speak about literature. There is a large amount of speculative interpretations, i.e. interpretations which are based on secondary sources (a researcher finds a correlation between reward dependence and positive emotionality. Literature has documented a correlation between positive emotionality and social adjustment. The researcher writes that his or her findings have shown an association between RD and adjustment). The discussion also includes statements derived from concepts which have not been used in the present study. The discussion should be a more coherent wholeness; in its present form it is more likely to comprise a long list of random correlations, most of those correlations being well-known, not novel. A significance of those associations would need more clarification. At the end of the discussion there are two slightly strange suggestions. First, the authors suggest that instead of self-reports, one’s personality could be assessed by others, teachers or peers. This is not possible, then it is no more one’s personality but observation on his or her behavior. Second, the authors suggest that physiological measures like heart rate variability could be used to understand relations between TCI and subjective wellbeing

---

## Round 0.2 · accepted · Accept

· Academic Editor

Accept

I thought the revision process was very well handled and the paper was improved. Congratulations!